# MSTA3D: Multi-scale Twin-attention for 3D Instance Segmentation

## ABSTRACT

Recently, transformer-based techniques incorporating superpoints have become prevalent in 3D instance segmentation. However, they often encounter an over-segmentation problem, especially noticeable with large objects. Additionally, unreliable mask predictions stemming from superpoint mask prediction further compound this issue. To address these challenges, we propose a novel framework called MSTA3D. It leverages multi-scale feature representation and introduces a twin-attention mechanism to effectively capture them. Furthermore, MSTA3D integrates a box query with a box regularizer, offering a complementary spatial constraint alongside semantic queries. Experimental evaluations on ScanNetV2, ScanNet200 and S3DIS datasets demonstrate that our approach surpasses state-of-the-art 3D instance segmentation methods. The code will be released upon paper publication.

## CCS CONCEPTS

• **Computing methodologies → Scene understanding**;

## KEYWORDS

Instance segmentation, 3D point cloud instance segmentation, vision transformer, multi-scale feature representation

## 1 INTRODUCTION

Given 3D point clouds, 3D instance segmentation refers to a task that involves identifying and separating individual objects within a 3D scene, including detecting object boundaries and assigning a unique label to each identified object. Its significant role in computer vision has surged corresponding to the demand for 3D perception in various applications, such as augmented/virtual reality [16, 24], autonomous driving [37], robotics [35], and indoor scanning [17].

In the literature, 3D instance segmentation methods are commonly categorized into four main approaches: proposal-based [5, 10, 12, 19, 26, 36], grouping-based [3, 11, 18, 32, 33, 39], kernel-based [8, 9, 11, 23, 34], and transformer-based methods [14, 21, 29, 30]. Proposal-based methods begin by generating a 3D bounding box and then using it to segment into an instance mask. Grouping-based methods aggregate points into clusters using per-point features, such as semantic or geometric cues, and then segment instances based on these clusters. Kernel-based methods are similar to grouping-based techniques but treat each potential instance as a

**Unpublished working draft. Not for distribution.**

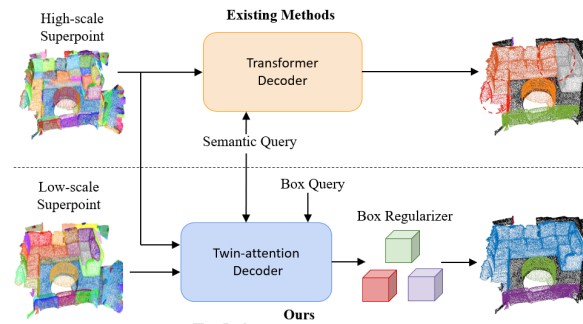

**Figure 1: The proposed MSAT3D, a 3D instance segmentation framework, tackles existing challenges by leveraging multi-scale feature representation and spatial query/regularizer.**

kernel for dynamic convolution. However, these methods require a high-quality proposal or clustering algorithm that heavily relies on per-point prediction, resulting in a significant demand for computational resources.

To address these issues, transformer-based methods have been proposed, treating each potential instance as an instance query and refining it through a series of transformer decoder blocks. However, predicting instances from point clouds inherently presents substantial challenges due to their typically lacking clear structure, unlike the regular grid arrangement found in images. Moreover, managing large-scale input point clouds further requires costly computations and extensive memory resources. Thus, recent transformer approaches leverage superpoints, which roughly offer contextual relationships between object parts with reduced memory usage. Nevertheless, existing transformer-based approaches employing superpoints often suffer from performance degradation due to over-segmentation problems and the unreliability of mask predictions. These challenges are described as follows: (1) Existing methods are prone to over-segmentation, especially when dealing with large objects such as doors, curtains, bookshelves, and backgrounds. Additionally, converting labels from superpoints to points can introduce unreliability into the categorical grouping. (2) The learning process for point-wise classification and aggregation encounters challenges due to the sparse and irregular distribution of observed scene points.

Hence, we propose a novel framework that leverages multi-scale superpoint features and simultaneously incorporates global/local spatial constraints. This framework is aimed at mitigating previously mentioned over-segmentation challenges and overcoming the limitations in point-wise classification. Specifically, to capture features at various scales, we generate superpoints at different scales, enabling effective feature representation of large objects and backgrounds as well as small objects. Correspondingly, we introduce a novel attention scheme, named twin-attention, to effectively fuse features at different scales. Moreover, we introduce the concept of

box query, in addition to semantic query, which is trained using the proposed twin-attention decoder and refined by the spatial regularizer to enhance the confidence of mask predictions. Furthermore, we utilize this bounding box prediction to enhance the reliability of mask predictions during the inference phase.

In summary, the contributions of this study are as follows:

- We propose a twin-attention-based decoder for effectively representing multi-scale features to tackle over-segmentation challenges observed in large objects and backgrounds.
- We introduce the notion of box query with box regularizer to provide complementary supervision without additional annotation requirements. This enforces spatial constraints for each instance during the query learning process, resulting in enhanced object localization and reduced background noise.
- Extensive experiments are conducted on widely-used benchmark datasets, including ScanNetV2 [4], ScanNet200 [28], and S3DIS [1], demonstrating the effectiveness of the proposed approach and achieving state-of-the-art results.

## 2 RELATED WORK

Current approaches of instance segmentation on 3D point clouds can be classified into four categories: proposal-based, grouping-based, kernel-based, and transformer-based methods.

**Proposal-based methods.** In the early stages of this approach, 3D-BoNet [36] was introduced, employing global characteristics derived from PointNet++ [26] to generate bounding boxes. These bounding boxes were then integrated with point features to produce instance masks. GICN [19] utilized a Gaussian distribution to estimate the center and size of each object instance for proposal prediction. Meanwhile, 3D-MPA [5] predicted instance centers and employed a graph convolutional network to group points around these centers, refining the features of proposals.

**Grouping-based methods.** SSTNet [18] represented the scene comprehensively by constructing a superpoint tree and traversing it to merge nodes, thus creating instance masks. PointGroup [11] predicted the 3D displacement of each point from its instance's center and identified clusters from both the original and center-shifted points. HAIS [3] introduced a hierarchical clustering technique, allowing smaller clusters to be either eliminated or merged into larger ones. SoftGroup [32] permitted each point to be associated with multiple clusters representing diverse semantic classes to mitigate prediction inaccuracies. Softgroup++ [33] extended SoftGroup to reduce computation time and search space for the clustering process.

**Kernel-based methods.** DyCo3D [8] employed a clustering algorithm from PointGroup [11] and lightweight 3D-UNet to generate kernels, while PointInst3D [9] opted for farthest-point sampling to produce kernels. DKNet [34] introduced candidate localization to yield more distinctive instance kernels. ISBNet [23] proposed a sampling-based instance-wise encoder to obtain a faster and more robust kernel for dynamic convolution.

**Transformer-based methods.** SPFormer [30] and Mask3D [29] utilized the mask-attention to produce query instances based on voxel or superpoint features while MAFT [14] improved efficiency by using position query instead of mask-attention. QueryFormer [21] optimized query instances with the query initialization module and proposes an affiliated transformer decoder to eliminate noise background.

## 3 BOX GUIDED TWIN-ATTENTION DECODER FOR MULTI-SCALE SUPERPOINT

In this section, we provide an in-depth discussion of the design choices underlying the proposed model. The problem addressed in this paper and the overall structure devised to tackle this problem are elucidated in Section 3.1. Section 3.2 explains the backbone network and the reasoning behind incorporating multi-scale superpoint features. Section 3.3 elaborates on the novel twin attention mechanism, and Section 3.4 outlines the approach for constraining spatial regions for each instance. Finally, Section 3.5 provides a comprehensive explanation of each component of the loss function and the matching method employed.

### 3.1 Architecture Overview

The goal of 3D instance segmentation in this paper is to predict precise point-wise boundaries of $N_I$ individual object instances given $N_h$ superpoints generated from point clouds, denoted as $\mathbf{p} \in \mathbb{R}^{N \times 6}$. Each point cloud contains positional coordinates $(x, y, z)$ and color information $(r, g, b)$, where $N$ is the total number of point clouds. A binary mask represents each of these $N_I$ instances, and a set of binary masks is collectively referred to $\mathbf{M} \in \{0, 1\}^{N_I \times N_h}$, where a value of 1 indicates the presence of objects, and 0 indicates their absence. Moreover, it is necessary to predict the semantic category associated with each instance, denoted as $\mathbf{C} \in \mathbb{Z}^{N_I \times N_C}$, where $N_C$ is the total number of semantic categories.

To tackle the problem, we propose a model consisting of three key components, as illustrated in Figure 2: (1) backbone network, (2) twin-attention decoder, and (3) box regularizer. First, the backbone network extracts multi-scale features $\mathbf{S}_\ell$ and $\mathbf{S}_h$ from the given inputs. These extracted features are then fed into the novel twin-attention decoder to generate instance proposals, represented as $\mathbf{X}$, guided by newly introduced box queries. Additionally, employing the box regularizer, the prediction of bounding boxes $\widetilde{\mathbf{B}}$ along with their corresponding scores $\widetilde{\mathbf{B}}_s$ are utilized to confine instance regions. Finally, through the twin-attention-based decoder and box regularizer, the final outputs include instance masks $\widetilde{\mathbf{M}}$ and corresponding labels $\widetilde{\mathbf{C}}$.

### 3.2 Backbone Network

As previously mentioned in Section 3.1, the backbone network, a 3D U-Net [6, 27], takes voxelized point clouds as input and extracts point-wise features. The voxelization method utilized in this paper is based on the approach outlined in [14, 21, 30].

We choose to utilize superpoint-wise labels instead of point-wise labels to reduce training time and memory consumption (See Figure 5). Superpoints, as proposed by [15], comprise multiple points sharing similar geometric properties and serve as a method to subsample point clouds. However, one potential drawback of using superpoints is the possibility of over-clustering scene [18, 21, 23, 30], which depends on the chosen grouping size. Therefore, we extract features from superpoints at multiple scales to ensure that the

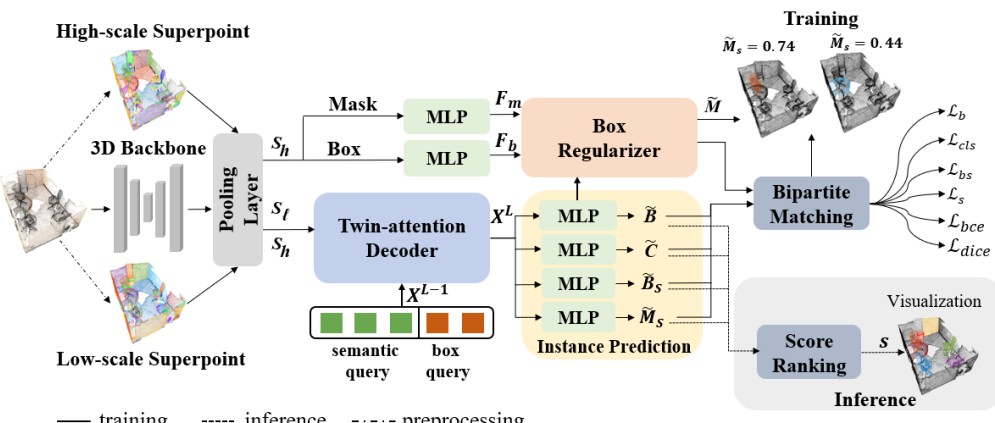

**Figure 2: The MSTA3D framework for instance segmentation on point clouds.**

decoder accurately identifies objects of various sizes during the learning process.

For this purpose, we pre-compute superpoint at two different scales by adjusting the number of neighbors for each point, as described in [15]. Following this, a pooling layer is applied to aggregate information from the point-wise features into the multi-scale features. We then denote the outputs passing through this pooling layer as $S_\ell \in \mathbb{R}^{N_\ell \times D_b}$ for the lower-scale features and $S_h \in \mathbb{R}^{N_h \times D_b}$ for the higher-scale features. Here, $N_\ell$ and $N_h$ represent the number of low and high-scale superpoints, respectively, and $D_b$ is the embedding dimension from the backbone.

## 3.3 Twin-attention Decoder

To leverage the advantages of multi-scale features $S_\ell$ and $S_h$ defined in Section 3.2, we introduce a novel decoder structure consisting of a series of twin attentions. The proposed twin-attention-based decoder is meticulously designed to integrate the multi-scale features. This decision is driven by the need to ensure that the proposed model effectively harnesses the diverse information present across different scales of the input features. The twin decoder incorporates a spatial regularizer for guiding instance delineations. Further details regarding this spatial regularization are provided in the following section.

**Region Constraint Instance Query.** We propose the concept of a box query, along with a semantic query [14, 21, 30], to guide the regions of instance masks for more accurate predictions. This guidance enables the model to concentrate more effectively on regions of interest, potentially reducing instance region ambiguity and improving segmentation accuracy. Concretely, we construct a set of learnable instance queries $X^0 \in \mathbb{R}^{N_o \times D_o}$ by concatenating the box queries with 6 elements, denoted as $X_b \in \mathbb{R}^{N_o \times 6}$ and the semantic queries with $D_s$ elements, denoted as $X_s \in \mathbb{R}^{N_o \times D_s}$. The concatenation is expressed as $X^0 = [X_s; X_b]$, where $N_o$ denotes the number of queries (proposals), initially set randomly with $N_o > N_I$, and $D_o$ is the total dimensionality of a query vector (i.e., $D_o = D_s + 6$). Note that the number of instance proposals $N_o$ is configured to be larger than the ground truth $N_I$. Consequently, the proposals selected as the final ones are those with the highest confidence.

**Twin-Attention-Based Feature Extraction.** The proposed twin-attention-based decoder is structured to concurrently process low and high-scale features through weight-shared attention layers, as illustrated in Figure 3. It consists of a stack of six twin-attention blocks, indexed by $L$ ($L = 1, \cdots, 6$), to process the outputs. Each block includes three sub-modules: cross and self-attention, feature fusion, and an instance prediction module.

Let $\pi_c(\cdot)$ be the linear projection in the attention modules, where $c = \{q, v, k\}$ represents query, value, and key, respectively. In the cross-attention modules, the query matrix $Q^L \in \mathbb{R}^{N_o \times D_o}$ of the $L$-th block is obtained after linearly projecting instance queries, computed as $\pi_q^L\left(X^{L-1}\right)$. Similarly, for low-scale feature $S_\ell$, the key matrix $K_\ell^L \in \mathbb{R}^{N_\ell \times D_o}$ and value matrix $V_\ell^L \in \mathbb{R}^{N_\ell \times D_o}$ are given by linearly projecting $\pi_k^L(S_\ell)$ and $\pi_v^L(S_\ell)$, respectively. Correspondingly, the key and value matrices of the high-scale features $S_h$ are derived as $K_h^L = \pi_k^L(S_h) \in \mathbb{R}^{N_h \times D_o}$ and $V_h^L = \pi_v^L(S_h) \in \mathbb{R}^{N_h \times D_o}$, respectively, using the same procedure. The proposed twin attention (TATT) is then computed on the shared set of instance queries and multi-scale features to attend to relevant information, as follows:

$$TATT(Q^L, K_h^L, V_h^L) = \sigma\left(\frac{Q^L\left(K_h^L\right)^T}{\sqrt{d}} + A_h^{L-1}\right) \cdot V_h^L, \quad (1a)$$

$$TATT(Q^L, K_\ell^L, V_\ell^L) = \sigma\left(\frac{Q^L\left(K_\ell^L\right)^T}{\sqrt{d}}\right) \cdot V_\ell^L, \quad (1b)$$

where $\sigma(\cdot)$ denotes the softmax function, and $d$ is computed as the division of the dimension of the query ($D_o$) by the number of attention heads (8 in the experiments of this study). For high-scale attention in (1a), we employ the superpoint mask attention $A_h^{L-1} \in \mathbb{R}^{N_o \times N_h}$ with a threshold $\tau = 0.5$, similar to the works in [21, 29, 30]. However, we further enhance the mask attention prediction by incorporating the proposed box query and box regularizer, as elaborated in Section 3.4.

Next, denoting the outputs of the cross-attention modules passing through a residual connection [7] and layer norm [2, 31] as

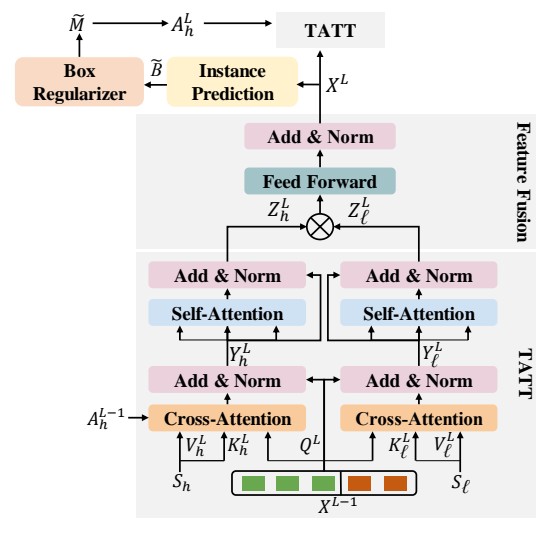

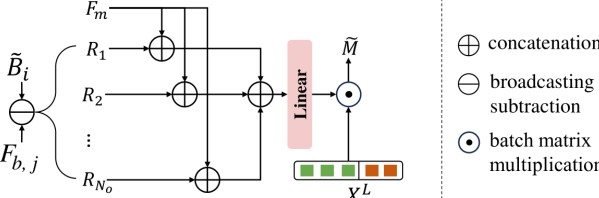

Figure 3: The architecture of twin-attention-based decoder. The twin-attention-based decoder fuses multi-scale features $S_h$ and $S_\ell$ and predicts $X^L$ by refining box queries.

$Y_\ell^L \in \mathbb{R}^{N_o \times D_o}$ and $Y_h^L \in \mathbb{R}^{N_o \times D_o}$, the subsequent self-attention layer computes twin attention on a set of linearly projected queries, keys, and values, in a similar manner to (1a) and (1b), without applying mask attention. The computed outputs of the self-attention layer, denoted as $Z_\ell^L \in \mathbb{R}^{N_o \times D_o}$ and $Z_h^L \in \mathbb{R}^{N_o \times D_o}$, are then fed into the multi-scale feature fusion module to abstract both low-scale and high-scale contextual information. To accomplish this, we utilize element-wise multiplication to merge the features extracted from inputs of different scales, followed by further aggregation using a feedforward layer (FFN), as follows:

$$X^L = FFN\left(Z_\ell^L \otimes Z_h^L\right), \qquad (2)$$

where $\otimes$ denotes element-wise multiplication. The output $X^L \in \mathbb{R}^{N_o \times D_o}$ of this feature fusion module for $L$-th block is then utilized for the inputs of the next twin-attention blocks and box regularizer. In the training, the proposed twin-attention blocks are sequentially trained using an iterative prediction strategy [30]. The output of the last decoder block ($X^6$) serves as the final instance proposals during inference.

We have found that this introduced twin-attention-based decoder indeed enhances the performance of instance segmentation by leveraging multi-scale features. It enables the representation of objects of various sizes and captures the background details of the entire scene, effectively mitigating the possibility of over-segmentation (See Table 6 and Figure 6).

**Instance and Box Prediction:** Given the fused feature output $X^L$ of the $L$-th block, the instance prediction module computes the mask score $\widetilde{M}_s \in \mathbb{R}^{N_o}$ for each instance and its corresponding class $\widetilde{C} \in \mathbb{R}^{N_o \times N_C}$ through multilayer perceptron (MLP) layers. These MLP layers employ linear activation for the mask score and softmax activation for classification. Moreover, to fully exploit the information from latent instance queries, we introduce two additional regressors with linear activation: $\widetilde{B} \in \mathbb{R}^{N_o \times 6}$ of instance box predictions and the corresponding box scores $\widetilde{B}_s \in \mathbb{R}^{N_o}$.

Figure 4: The architecture of box regularizer. The box regularizer predicts positional differences between bounding boxes derived from scene-wise features and those derived from instance-wise features.

These spatial feature-associated regressors are introduced without requiring additional annotation efforts. The proposed framework autonomously derives box information from the instance labels. Specifically, we choose axis-aligned bounding boxes due to the simplicity of constructing ground truths $B \in \mathbb{R}^{N_I \times 6}$ from existing instance annotations. Each bounding box is defined by six coordinates, representing the minimum and maximum 3D coordinates (*i.e.*, $[x_{min}, y_{min}, z_{min}, x_{max}, y_{max}, z_{max}]^T$) that enclose the instance. We observe that adopting this approach improves the overall performance of the proposed method because it utilizes bounding boxes as local spatial references for the box regularizer. These references provide contextual information about the local space occupied by each object relative to the entire scene during training, helping the model generate high-quality superpoint masks.

## 3.4 Spatial Constraint Regularizer

In addition to incorporating box queries, we introduce the box regularizer to identify coarse object regions and constrain local spatiality. This regularization aims to enhance the representation of spatial latent features, utilizing box queries passed through the proposed twin-attention, as demonstrated in Table 6.

The proposed box regularizer takes the box predictions $\widetilde{B}$ for each instance from the twin-attention-based decoder as input, along with features comprising scene-wise semantic score $F_m \in \mathbb{R}^{N_h \times D_s}$ for each superpoint and scene-wise box information $F_b \in \mathbb{R}^{N_h \times 6}$. These scene-wise features are obtained by passing $S_h$ through multilayer perceptron layers with linear activation. By utilizing these features, the goal is to provide the box regularizer with comprehensive scene-associated features, ensuring effective representation of local instance regions. This approach guarantees the enhanced confidence of superpoint mask predictions by analyzing instance features in the context of the entire scene. Concretely, the box regularizer integrates instance-specific positional features $\widetilde{b}_i \in \mathbb{R}^6$, which denotes $i$-th row of $\widetilde{B}$, and scene-wise positional features $F_b$ for all instances. By employing the broadcasting subtraction operator, we compute the relative positional difference $R_i \in \mathbb{R}^{N_h \times 6}$ for each instance box in relation to the entire scene, as follows:

$$R_i = \{\widetilde{b}_i\}^{N_h} \ominus F_b, \qquad (3)$$

where $\ominus$ denotes a broadcasting subtraction operator, and $\{\widetilde{b}_i\}^{N_h}$ represents the stretched version of $\widetilde{b}_i$ by $N_h$.

To predict the binary mask in each twin-attention block, we concatenate these relative positional features $R_i$ with the scene-wise semantic features $F_m \in \mathbb{R}^{N_h \times D_s}$ to generate new $N_o$ features

that simultaneously capture spatial and semantic information. The predicted binary masks are computed through batch matrix multiplication, as follows:

$$\widetilde{\mathbf{M}} = Linear\left(\left[\{\mathbf{R}_i\}_{i=1}^{N_o}; \{\mathbf{F}_m\}^{N_o}\right]\right) \odot \mathbf{X}^L. \quad (4)$$

In (4), $\{\mathbf{R}_i\}_{i=1}^{N_o}$ constructs to a tensor in $\mathbb{R}^{N_o \times (N_h \times 6)}$, and $\{\mathbf{F}_m\}^{N_o}$ represents the stretched version of $\{\mathbf{F}_m\}$ by $N_o$ using broadcasting operators, resulting in the concatenated features in $\mathbb{R}^{N_o \times (N_h \times D_o)}$ (where $D_o = D_s + 6$); $\odot$ denotes batch matrix multiplication, and $Linear(\cdot)$ denotes a linear activation function.

Intuitively, the relative positional variation $\mathbf{R}_i$ captures the prediction differences between the bounding box predicted from scene-wise global dependency and the estimated box derived from instance-wise local dependency. This suggests that mask prediction can be enhanced by abstracting both scene-wise features and individual instance-wise features. Moreover, this prediction of relative positional disparities can offer valuable feedback to the model, enabling it to refine its predictions based on discrepancies observed at different scales.

## 3.5 Training and Inference

**Training:** To train the proposed framework, we formulate the loss function $\mathcal{L}$, as follows:

$$\mathcal{L} = \beta_{cls}\cdot\mathcal{L}_{cls} + \beta_{mask}\cdot(\mathcal{L}_{bce}+\mathcal{L}_{dice}) + \beta_s\cdot\mathcal{L}_s + \beta_b\cdot\mathcal{L}_b + \beta_{bs}\cdot\mathcal{L}_{bs} \quad (5)$$

where the coefficients $\beta_{cls}$, $\beta_s$, $\beta_{mask}$, $\beta_{bs}$, and $\beta_b$ determine the contribution of each loss function to the total loss function. The classification loss $\mathcal{L}_{cls}$ is defined by the multi-class cross-entropy loss for object class categorization, and mask prediction utilizes a combination of binary cross-entropy $\mathcal{L}_{bce}$ and dice loss $\mathcal{L}_{dice}$. Additionally, we use L2 loss $\mathcal{L}_s$ for predicting mask scores used for matching during training, L1 loss $\mathcal{L}_b$ for box coordinate prediction, and L2 loss $\mathcal{L}_{bs}$ for box score prediction, as follows:

$$\mathcal{L}_s = \frac{1}{\|\mathbb{1}_{iou_{ms}}\|_1} \cdot \left\|\mathbb{1}_{iou_{ms}} \otimes \left(\widetilde{\mathbf{M}}_s - iou_{ms}\right)\right\|_2^2, \quad (6a)$$

$$\mathcal{L}_b = \frac{1}{N_I}\left\|\widetilde{\mathbf{B}} - \mathbf{B}\right\|_{1,1}, \quad (6b)$$

$$\mathcal{L}_{bs} = \frac{1}{\|\mathbb{1}_{iou_{bs}}\|_1} \cdot \left\|\mathbb{1}_{iou_{bs}} \otimes \left(\widetilde{\mathbf{B}}_s - iou_{bs}\right)\right\|_2^2, \quad (6c)$$

where $\|\cdot\|_p$ denotes the vector $p$-norm, and $\|\cdot\|_{p,q}$ denotes the entry-wise matrix $p, q$-norm; $\mathbb{1}_{\{iou_{ms}\}}$ indicates whether the Intersection over Union (IoU) between mask prediction and assigned ground truth is higher than a threshold $\eta_{ms}$; and $\mathbb{1}_{\{iou_{bs}\}}$ indicates whether IoU between box prediction and assigned ground truth is higher than a threshold $\eta_{bs}$. In the experiments, we used $\beta_{cls} = 0.5$ for class classification, $\beta_{mask} = 1$ for mask prediction, $\beta_s = 0.5$ for mask score prediction, $\beta_{bs} = 0.5$ for box score prediction, $\beta_b = 1$ for box prediction, and $\eta_{ms} = \eta_{bs} = 0.5$.

To assign proposals to each ground truth instance, we formulate a pairwise matching cost $C_{ij}$ to assess the similarity between the $i$-th proposal and the $j$-th ground truth. The matching cost $C_{ij}$ is defined as [30]:

$$C_{ij} = -\lambda_{cls}\cdot p_{i,c_j} + \lambda_{mask}C_{ij}^{mask}, \quad (7)$$

where $p_{i,c_j}$ denotes the $c_j$-th semantic category probability of the $i$-th proposal; $C_{ij}^{mask}$ denotes superpoint mask matching score; $\lambda_{cls}$ and $\lambda_{mask}$ are the coefficients of each term respectively. In the experiments, we set $\lambda_{cls} = 0.5$ and $\lambda_{mask} = 1$. The superpoint mask matching cost $C_{ij}^{mask}$ is calculated based on a binary cross-entropy (BCE) and a dice loss with a Laplace smoothing [22]:

$$C_{ij}^{mask} = BCE\left(\widetilde{\mathbf{m}}_i, \mathbf{m}_j\right) + \frac{\widetilde{\mathbf{m}}_i \cdot \mathbf{m}_j^T + 1}{\|\widetilde{\mathbf{m}}_i\|_1 + \|\mathbf{m}_j\|_1 + 1}, \quad (8)$$

where $\widetilde{\mathbf{m}}_i$ is the $i$-th row of the superpoint mask prediction $\widetilde{\mathbf{M}}$, and $\mathbf{m}_j$ is the $j$-th row of the ground truth mask $\mathbf{M}$. Additionally, we treat the task of matching proposals with ground truth as an optimal assignment problem. Hence, we employ the Hungarian algorithm [13] to identify the optimal solution for this task. Once completing the assignment, we categorize the proposals not assigned to ground truth as the "no instance" class.

**Inference.** During inference, we compute instance confidence, considering that the number of proposals $N_o$ can be greater than the number of ground truth instances $N_I$. The confidence score of $i$-th proposal is computed based on classification probability $\widetilde{\mathbf{c}}_i$, IoU box score $\widetilde{b}_{s,i}$, IoU score $\widetilde{m}_{s,i}$ of corresponding superpoint mask. Follow [14, 30], we utilize a superpoint mask score $a_i$ by computing by averaging the probabilities of superpoint that have a probability greater than 0.5 in each superpoint mask prediction. The final confidence score for each instance is defined as $\{s\}_{i=1}^{N_o} = \left\{\widetilde{\mathbf{c}}_i \cdot \widetilde{b}_{s,i} \cdot \widetilde{m}_{s,i} \cdot a_i\right\}_{i=0}^{N_o}$ and utilized for ranking predicted instances. In this work, we do not employ post-processing (non-maximum suppression or DBSCAN) as [21, 23, 29] to enhance inference speed.

## 4 EXPERIMENTAL RESULTS

In this section, we introduce the datasets and evaluation metrics utilized to validate the effectiveness of the proposed model in Section 4.1, and implementation details are described in Section 4.2. Following this, performance comparisons are presented in Section 4.3, and analyses of additional studies on the model are demonstrated in Section 4.4.

## 4.1 Dataset and Evaluation Metric

**Dataset:** To validate the effectiveness of the proposed framework, we conducted experiments on the ScanNetV2 dataset [4], ScanNet200 dataset [28], and S3DIS dataset [1].

ScanNetV2 comprises 1613 scans, with 1201 scans for training, 312 for validation, and 100 for testing. It includes 18 object categories commonly used for 3D instance segmentation evaluation.

ScanNet200 [28] extends ScanNetV2 [4] with fine-grained categories and includes 198 instances with an additional 2 semantic classes. The dataset division into training, validation, and testing sets mirrors the original ScanNetV2 dataset.

S3DIS consists of 271 scenes across 6 different areas, with each area containing 13 categories. For the experiments in this paper, evaluations were conducted using datasets of Area 5.

**Evaluation Metric:** The evaluation of instance segmentation typically relies on task-mean average precision (mAP), a widely-used metric that averages scores across various IoU thresholds ranging

**Table 1: Performance comparisons of 3D instance segmentation on ScanNetV2 hidden test set in terms of the mean average precision and mAP$_{25}$ scores for each class**

| Methods | Venue | mAP | mAP$_{50}$ | mAP$_{25}$ | bath | bed | bkshf | cabinet | chair | counter | curtain | desk | door | other | picture | fridge | s. cur. | sink | sofa | table | toilet | window |
|---|---|---|---|---|---|---|---|---|---|---|---|---|---|---|---|---|---|---|---|---|---|---|
| DyCo3D [8] | CVPR 21 | 39.5 | 64.1 | 76.1 | 100 | 93.5 | 89.3 | 75.2 | 86.3 | 60.0 | 58.8 | 74.2 | 64.1 | 63.3 | 54.6 | 55.0 | 85.7 | 78.9 | 85.3 | 76.2 | 98.7 | 69.9 |
| SSTNet [18] | ICCV 21 | 50.6 | 78.9 | 69.8 | 100 | 84.0 | 88.8 | 71.7 | 83.5 | 71.7 | 68.4 | 62.7 | 72.4 | 65.2 | 72.7 | 60.0 | 100 | 91.2 | 82.2 | 75.7 | 100 | 69.1 |
| HAIS [3] | ICCV 21 | 45.7 | 69.9 | 80.3 | 100 | 99.4 | 82.0 | 75.9 | 85.5 | 55.4 | 88.2 | 82.7 | 61.5 | 67.6 | 63.8 | 64.6 | 100 | 91.2 | 79.7 | 76.7 | 99.4 | 72.6 |
| DKNet [34] | ECCV 22 | 53.2 | 71.8 | 81.5 | 100 | 93.0 | 84.4 | 76.5 | 91.5 | 53.4 | 80.5 | 80.5 | 80.7 | 65.4 | 76.3 | 65.0 | 100 | 79.4 | 88.1 | 76.6 | 100 | 75.8 |
| PBNet [38] | ICCV 23 | 57.3 | 74.7 | 82.5 | 100 | 96.3 | 83.7 | 84.3 | 86.5 | 82.2 | 64.7 | 87.8 | 73.3 | 63.9 | 68.3 | 65.0 | 100 | 85.3 | 87.0 | 82.0 | 100 | 74.4 |
| TD3D [12] | WACV 24 | 48.9 | 75.1 | 87.5 | 100 | 97.6 | 87.7 | 78.3 | 97.0 | 88.9 | 82.8 | 94.5 | 80.3 | 71.3 | 72.0 | 70.9 | 100 | 93.6 | 93.4 | 87.3 | 100 | 79.1 |
| SoftGroup [32] | CVPR 22 | 50.4 | 76.1 | 86.5 | 100 | 96.9 | 86.0 | 86.0 | 91.3 | 55.8 | 89.9 | 91.1 | 76.0 | 82.8 | 73.6 | 80.2 | 98.1 | 91.9 | 87.5 | 87.7 | 100 | 82.0 |
| ISBNet [23] | CVPR 23 | 55.9 | 75.7 | 83.5 | 100 | 95.0 | 73.1 | 81.9 | 91.8 | 79.0 | 74.0 | 85.1 | 83.1 | 66.1 | 74.2 | 65.0 | 100 | 93.7 | 81.4 | 83.6 | 100 | 76.5 |
| SPFormer [30] | AAAI 23 | 54.9 | 77.0 | 85.1 | 100 | 99.4 | 80.6 | 77.4 | 94.2 | 63.7 | 84.9 | 85.9 | 88.9 | 72.0 | 73.0 | 66.5 | 100 | 91.1 | 86.8 | 87.3 | 100 | 79.6 |
| Mask3D [29] | ICRA 23 | 55.2 | 78.0 | 87.0 | 100 | 98.5 | 78.2 | 81.8 | 93.8 | 76.0 | 74.9 | 92.3 | 87.7 | 76.0 | 78.5 | 82.0 | 100 | 91.2 | 86.4 | 87.8 | 98.3 | 82.5 |
| MAFT [14] | ICCV 23 | **59.6** | 78.6 | 86.0 | 100 | 99.0 | 81.0 | 82.9 | 94.9 | 80.9 | 68.8 | 83.6 | 90.4 | 75.1 | 76.6 | 74.1 | 100 | 86.4 | 84.8 | 83.7 | 100 | 82.8 |
| QueryFormer [21] | ICCV 23 | 58.3 | 78.7 | 87.4 | 100 | 97.8 | 80.9 | 87.6 | 93.6 | 70.2 | 71.6 | 92.0 | 87.5 | 76.6 | 77.2 | 81.8 | 100 | 99.5 | 91.6 | 89.2 | 100 | 76.7 |
| **MSTA3D (Ours)** | - | 56.9 | **79.5** | **87.9** | 100 | 99.4 | 92.1 | 80.7 | 93.9 | 77.1 | 88.7 | 92.3 | 86.2 | 72.2 | 76.8 | 75.6 | 100 | 91.0 | 90.4 | 83.6 | 99.9 | 82.4 |

**Table 2: Performance comparisons of 3D instance segmentation on ScanNetV2 validation set in terms of mean average precision (mAP).**

| Methods | mAP | mAP$_{50}$ | mAP$_{25}$ |
|---|---|---|---|
| DyCo3D [8] | 35.4 | 57.6 | 72.9 |
| HAIS [3] | 43.5 | 64.4 | 75.6 |
| DKNet [34] | 50.8 | 66.7 | 76.9 |
| SoftGroup [32] | 45.8 | 67.6 | 78.9 |
| PBNet [38] | 54.3 | 70.5 | 78.9 |
| TD3D [12] | 47.3 | 71.2 | 81.9 |
| ISBNet [23] | 54.5 | 73.1 | 82.5 |
| Mask3D [29] | 55.2 | 73.7 | 83.5 |
| SPFormer [30] | 56.3 | 73.9 | 82.9 |
| QueryFormer [21] | 56.5 | 74.2 | 83.3 |
| MAFT [14] | 58.4 | 75.6 | 84.5 |
| **MSTA3D (Ours)** | **58.4** | **77.0** | **85.4** |

from 50% to 95%, incremented by 5%. mAP$_{50}$ and mAP$_{25}$ represent the scores at IoU thresholds of 50% and 25%, respectively. Our evaluation of the ScanNetV2 and ScanNet200 datasets included mAP, mAP$_{50}$, and mAP$_{25}$ metrics, and we also utilized mean precision (mPrec) and mean recall (mRec) in our analysis of the S3DIS dataset.

## 4.2 Implementation Details

We implemented the proposed model using the PyTorch deep learning framework [25] and conducted training with the AdamW optimizer [20]. The training was performed on a single A100 GPU with 40GB of memory. We used an initial learning rate of 0.0001, weight decay of 0.05, and batch size of 2, and employed a polynomial scheduler with a base of 0.9 for 512 epochs. For data augmentation, we applied random scaling, elastic distortion, random rotation, horizontal flipping around the z-axis, and random scaling for $x$, $y$, and $z$ coordinates. Additionally, we utilized noise and brightness augmentation for red, green, and blue colors. Both ScanNetV2 and ScanNet200 datasets were processed with a voxel size of $2cm$. For S3DIS dataset, we used a voxel size of $5cm$. The proposed model adopted the same backbone design described in [14, 30], resulting in a feature map with 32 channels ($D_b = 32$). During training, scenes were limited to a maximum of $250,000$ points. During inference, the entire scenes were inputted into the network without any cropping, and the top 100 instances were selected based on their highest scores.

## 4.3 Experimental Results

**ScanNetV2.** The instance segmentation results of the ScanNetV2 test and validation sets are presented in Tables 1[1] and 2. Overall, the proposed model shows a significant improvement in mAP compared to previous studies, indicating its capability to capture intricate details and generate high-quality instance segmentation. In particular, compared to SPFormer [30], the proposed model achieved a performance increase of +2.0 mAP, +2.5 mAP$_{50}$, and +2.8 mAP$_{25}$. Compared to Mask3D [29], the improvement was +1.7 mAP, +1.5 mAP$_{50}$, and +0.9 mAP$_{25}$. Compared to MAFT [14], the gains were +0.9 mAP$_{50}$ and +1.9 mAP$_{25}$. Finally, compared to QueryFormer [21], the increases were +0.8 mAP$_{50}$ and +0.5 mAP$_{25}$ on the hidden test set. On the validation set, the proposed model outperformed other methods across all three metrics: mAP, mAP$_{25}$, and mAP$_{50}$. Specifically, compared to QueryFormer [21], the proposed model achieved a +1.9 mAP improvement, +2.8 mAP$_{50}$ improvement, and +2.1 mAP$_{25}$ improvement. Compared to MAFT [14], the gains were +1.4 mAP$_{50}$ and +0.9 mAP$_{25}$. As evidenced by the mAP scores for each class, the proposed model exhibited superior performance, particularly for relatively large objects such as beds or bookshelves. Notably, it established a considerable lead of over 10% mAP$_{25}$ specifically for the bookshelf category. This demonstrates that our proposed method effectively captures large objects to mitigate over-segmentation.

---

[1]Note that the results presented were obtained from the ScanNet benchmark on March 25, 2024

**Table 3: Performance comparisons of 3D on ScanNet200 validation set. The asterisk(*) indicates reproduced results.**

| Methods | mAP | mAP$_{50}$ | mAP$_{25}$ |
|---|---|---|---|
| PointGroup [11] | - | 24.5 | - |
| PointGroup + LGround [28] | - | 26.1 | - |
| ISBNet [23] | 24.5 | 32.7 | - |
| SPFormer* [30] | 25.2 | 33.8 | 39.6 |
| TD3D* [12] | 23.1 | 34.8 | **40.4** |
| **MSTA3D (Ours)** | **26.2** | **35.2** | 40.1 |

**Table 4: Performance comparisons of 3D instance segmentation on Area 5 of the S3DIS dataset**

| Methods | mAP$_{50}$ | mPrec | mRec |
|---|---|---|---|
| DyCo3D [8] | - | 64.3 | 64.2 |
| DKNet [34] | - | 70.8 | 65.3 |
| HAIS [3] | - | 71.1 | 65.0 |
| SoftGroup [32] | 66.1 | 73.6 | 66.6 |
| TD3D [12] | 65.1 | 74.4 | 64.8 |
| PBNet [38] | 66.4 | 74.9 | 65.4 |
| SPFormer [30] | 66.8 | 72.8 | 67.1 |
| MAFT [14] | 69.1 | - | - |
| QueryFormer [21] | 69.9 | 70.5 | **72.2** |
| Mask3D [29] | **71.9** | 74.3 | 63.7 |
| **MSTA3D (Ours)** | 70.0 | **80.6** | 70.1 |

**ScanNet200.** We also evaluated the proposed model on the Scan-Net200 dataset using the validation set. As shown in Table 3, the results of the proposed model indicate a significant improvement compared to other methods. Specifically, the proposed model achieved an increase of +1 in mAP, +1.4 in mAP$_{50}$ and +0.5 in mAP$_{25}$ compared to SPFormer [30]. Especially, compared to TD3D [12], our model achieved a +3.1 mAP and +0.4 mAP$_{50}$ improvement. Nevertheless, this dataset presents challenges for superpoint-wise labels due to its limited number of classes, indicating that point-wise methods will likely remain dominant.

**S3DIS.** We conducted an evaluation of the proposed model on S3DIS using Area 5. As shown in Table 4, the results demonstrate slight improvements compared to previous works such as MAFT or QueryFormer. In comparison to SPFormer, we achieved a notable improvement of +3.2 mAP$_{50}$. Additionally, the proposed method outperformed other methods in terms of the mPrec metric.

### 4.4 Ablation Study

In this section, we conduct a thorough analysis to validate each component of the proposed approach, as outlined below.

**Multi-scale Feature Representation and Loss Functions.** We assessed performance to further analyze the effectiveness of multi-scale feature representations and the proposed loss functions by disabling one or more components. The results of this study are tabulated in Table 6. The results when utilizing instance bounding boxes confirmed that the introduced box queries improved the accuracy of instance recognition. Furthermore, utilizing bounding box features with box scores enhanced recognition accuracy even further. The introduced box regularizer also improved the accuracy

**Table 5: Performance comparisons with varying the number of queries on the ScanNetV2 validation set**

| Methods | # Queries | mAP | mAP$_{50}$ | mAP$_{25}$ |
|---|---|---|---|---|
| SPFormer [30] | 100 | 54.2 | 72.4 | 82.8 |
| SPFormer [30] | 200 | 55.2 | 73.3 | 82.4 |
| SPFormer [30] | 400 | 56.3 | 73.9 | 82.9 |
| SPFormer [30] | 800 | 55.9 | 73.7 | 83.8 |
| MAFT [14] | 400 | 58.4 | 75.6 | 84.5 |
| MSTA3D (Ours) | 150 | 57.2 | 76.1 | 84.3 |
| MSTA3D (Ours) | 200 | 57.4 | 75.8 | 85.2 |
| MSTA3D (Ours) | 250 | 58.2 | 76.9 | 85.4 |
| **MSTA3D (Ours)** | 300 | 58.4 | **77.0** | 85.4 |
| MSTA3D (Ours) | 350 | 58.6 | 76.6 | 85.2 |
| MSTA3D (Ours) | 400 | 58.1 | 76.4 | 84.7 |
| MSTA3D (Ours) | 450 | **58.9** | 76.7 | 85.0 |

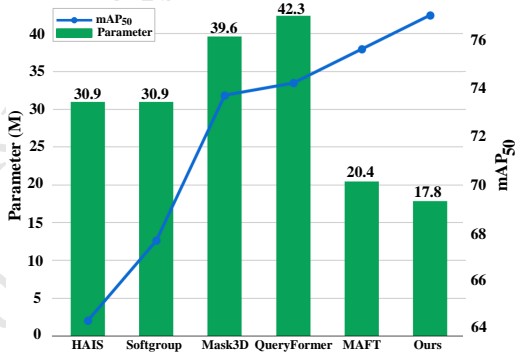

**Figure 5: Comparisons of model complexity**

of instance recognition, indicating the effectiveness of augmenting instance-wise spatial features with scene-wise spatial features from the superpoint-wise predictor.

When comparing the performance of multi-scale feature representation against single-scale feature representation, the results demonstrate that multi-scale superpoints enhanced the feature representation at different scales. This finding is consistent with the results presented in Table 1, where the proposed model outperformed relatively large objects while performing well in recognizing small objects.

**Number of Queries.** We conducted a performance comparison of the proposed model using varying numbers of queries as shown in Table 5. To ensure fairness in comparison, we exclusively evaluated models that utilize superpoint-wise information for both training and inference, namely SPFormer [30] and MAFT [14]. Remarkably, the proposed model achieved superior performance across all three metrics with only 150 queries, compared to SPFormer's 400 queries. Furthermore, the proposed model outperformed MAFT in two metrics, mAP, and mAP$_{50}$ with only 200 queries, and achieved better results in three metrics with 250 queries.

**Model Complexity.** To validate the efficiency of the proposed model alongside the analysis of the number of queries, we compared the number of parameters of various methods. The results

**Table 6: Ablation study on different loss functions using both single-scale feature and the proposed multi-scale feature representation**

| Instance Box | Box Score | Box Regularizer | Low-scale only | | | High-scale only | | | Multi-scale | | |
|---|---|---|---|---|---|---|---|---|---|---|---|
| | | | mAP | mAP$_{50}$ | mAP$_{25}$ | mAP | mAP$_{50}$ | mAP$_{25}$ | mAP | mAP$_{50}$ | mAP$_{25}$ |
| – | – | – | 47.2 | 67.4 | 78.0 | 56.3 | 73.9 | 82.9 | 56.4 | 75.0 | 84.0 |
| ✓ | – | – | 47.3 | 67.1 | 78.8 | 56.7 | 74.6 | **84.6** | 57.0 | 74.8 | 84.4 |
| – | – | ✓ | 48.0 | 68.0 | 79.6 | 56.2 | 74.6 | 83.5 | 56.9 | 75.5 | 84.3 |
| ✓ | ✓ | – | 48.3 | 68.6 | 79.4 | 56.8 | 75.6 | 83.8 | 57.1 | 75.2 | 84.3 |
| ✓ | – | ✓ | 48.1 | **68.9** | 79.9 | 57.4 | 75.0 | **84.6** | 57.5 | 75.4 | 84.8 |
| ✓ | ✓ | ✓ | **49.4** | 68.0 | **81.2** | **57.6** | **75.6** | 84.5 | **58.4** | **77.0** | **85.4** |

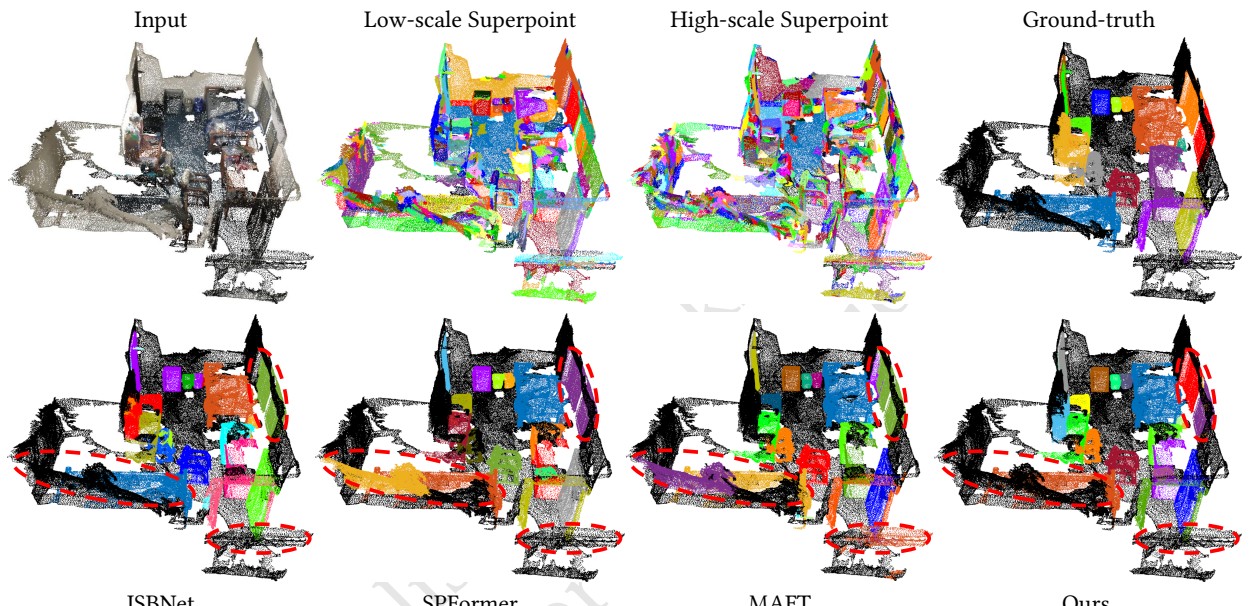

Figure 6: Qualitative comparison of the proposed model with other methods on ScanNetV2.

demonstrate that the proposed model achieved superior performance with significantly fewer parameters, specifically 24.5 million fewer than QueryFormer [21] and 2.6 million fewer than MAFT [14]. Compared to the recent transformer-based method Mask3D [29], the proposed model utilized fewer than 21.8 million parameters, as evidenced in Figure 5.

## 5 CONCLUSION

In this paper, we presented MSTA3D, a transformer-based method designed for 3D point cloud instance segmentation. To address the challenge of over-segmentation, particularly with background or large objects in the scene, we devised a multi-scale superpoint strategy. Furthermore, we introduced a twin-attention decoder to leverage both high-scale and low-scale superpoints simultaneously. This enhancement expands the model's capacity to capture features at various scales, thereby enabling better performance on large objects and reducing over-segmentation. In addition to the

semantic query, we introduced the notion of a box query. This provides spatial context for generating high-quality instance proposals, assists the box regularizer in producing reliable instance masks, and contributes to box score regression, leading to significant performance improvements. Finally, we rigorously evaluated each of these components through extensive experiments.

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
