# OpenReview forum: "Multi-scale Twin-attention for 3D Instance Segmentation"
_acmmm.org/ACMMM/2024/Conference — MM2024 Poster_

### Official Review · Reviewer_c9T5 · 2024-05-07

**Rating:** 2
**Confidence:** 2

**Summary:**

The author propose a twin-attention-based decoder for effectively representing multi-scale features to tackle over-segmentation challenges observed in large objects and backgrounds. Secondly, The author introduce the notion of box query with box regularizer to provide complementary supervision without additional annotation requirements. This enforces spatial constraints for each instance during the query learning process, resulting in enhanced object localization and reduced background noise.

**Strengths:**

The author has verified the feasibility of their method through a large number of experiments.

**Limitations:**

1. Twin attention was proposed and widely applied a few years ago. The author did not cite twin attention literature in the introduction section. We suggest that the author cite twin attention's work in appropriate places.
2. In the section 4.3, the author did not state why the mAP metric of our method in Table 1 is lower than that of the MAFT and QueryFormer methods.
3. In table 4, what is the reason why the MSTA3D method is lower than other comparison methods in both mAP50 and mRec methods?
4. We believe that the method proposed in this article has limitations in terms of novelty.

**Suitability:**

2

---

### Official Review · Reviewer_QSUq · 2024-05-24

**Rating:** 4
**Confidence:** 2

**Summary:**

This paper introduces a novel method for 3D instance segmentation that aims to address key issues such as the over-segmentation problem and unreliable mask predictions. To tackle these challenges, the paper proposes to combine multi-scale feature representation with a twin-attention mechanism and introduces a box regularizer.

**Strengths:**

- Extensive experiments have been conducted to demonstrate the effectiveness of the proposed method, achieving competitive results compared to existing baselines.
- This paper is well-written with a clear structure.

**Limitations:**

- As a key component of the method, further robustness experiments on the multi-scale feature representation could be provided. Specifically, it would be beneficial to explore how different hyperparameters, such as the number of neighbors mentioned in L254, affect the final performance of the method. Additionally, an investigation into the performance impacts of utilizing more levels of different scale features could provide deeper insights into the scalability of the approach.

- The paper shows inconsistent improvements, notably exhibiting inferior results (e.g., mAP50 and mRec) on the S3DIS dataset. Providing a detailed analysis of these results would be beneficial.

**Suitability:**

2

---

### Official Review · Reviewer_gtZr · 2024-05-26

**Rating:** 4
**Confidence:** 4

**Summary:**

This paper presents a new transformer-based 3D instance segmentation framework that utilizes multiscale superpoints for feature aggregation. A twin-attention mechanism takes input features from both high-scale superpoints and low-scale superpoints for better representation. Additionally, a box query is used to regularize semantic predictions. Experiments conducted on popular 3D indoor datasets (ScanNet-V2, ScanNet200, and S3DIS) show improved performance over existing methods.

**Strengths:**

•	The paper is generally well-written and easy to follow. The figures clearly illustrate the proposed framework.

•	The motivation for applying multiscale superpoints is straightforward and sensible.

•	The experiments are solid. This method achieves SOTA performance across different datasets. The ablation study demonstrates the effectiveness of each proposed component.

**Limitations:**

I have some questions need clarification.

•	The training of this network is conducted on a single 40G A100 GPU. How is the number of high-scale and low-scale superpoints controlled to avoid out-of-memory issues during training? As state in l.254, the authors adjust the number of neighbors to obtain different scales. The total number of superpoints can be very large in some scenarios (i.e. the theatre hall in S3DIS). There are some superpoint method can perform segmentation based on the given number of superpoints. Can the authors compare the pros and cons of these two types of superpoints generation strategies?

•	In Figure 5, the model complexity of the proposed method is lower than other methods. Where does this reduction likely come from? For example, is it due to a lighter backbone with fewer layers, a better-arranged network, or the elimination of some unnecessary components?

•	Can this network be trained from scratch, or does it rely on the pretrained backbone from other method?

•	There are a few relevant publications that are not cited. Since the total number of references in this paper is only 39, the authors might consider including more related works on grouping-based methods in Section 2.

[1] OccuSeg: Occupancy-aware 3D Instance Segmentation
[2] 3D Instance Segmentation via Multi-Task Metric Learning
[3] Learning Regional Purity for Instance Segmentation on 3D Point Clouds

**Suitability:**

3

---

### Meta-Review · Area_Chair_QNx9 · 2024-07-01

**Recommendation:** Accept (Poster)
**Confidence:** 5

**Metareview:**

This paper presents a new transformer-based 3D instance segmentation framework with three main novelties:
1) Multi-scale point feature representation
2) Box regularization integrating scene-wise (global) and instance-wise (local) feature fusion
3) Efficient model architecture optimization

This paper is well written and clarity.  The experiments are extensive and solid. The experimental results did not completely surpass the best results, but they are competitive overall considering the trade-off between accuracy and computational complexity.